# 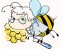 HoneyBee: Progressive Instruction Finetuning of Large Language Models for Materials Science

**Yu Song[1], Santiago Miret[2,*], Huan Zhang[3], Bang Liu[1,*†]**

[1]University of Montreal / Mila - Quebec AI, [2]Intel Labs [3]University of Waterloo
{yu.song, bang.liu}@umontreal.ca
santiago.miret@intel.com
h648zhan@uwaterloo.ca

## Abstract

We propose an instruction-based process for trustworthy data curation in materials science (MatSci-Instruct), which we then apply to finetune a LLaMa-based language model targeted for materials science (HoneyBee). MatSci-Instruct helps alleviate the scarcity of relevant, high-quality materials science textual data available in the open literature, and HoneyBee is the first billion-parameter language model specialized to materials science. In MatSci-Instruct we improve the trustworthiness of generated data by prompting multiple commercially available large language models for generation with an Instructor module (e.g. ChatGPT) and verification from an independent Verifier module (e.g. Claude). Using MatSci-Instruct, we construct a dataset of multiple tasks and measure the quality of our dataset along multiple dimensions, including accuracy against known facts, relevance to materials science, as well as completeness and reasonableness of the data. Moreover, we iteratively generate more targeted instructions and instruction-data in a finetuning-evaluation-feedback loop leading to progressively better performance for our finetuned HoneyBee models. Our evaluation on the MatSci-NLP benchmark shows HoneyBee's outperformance of existing language models on materials science tasks and iterative improvement in successive stages of instruction-data refinement. We study the quality of HoneyBee's language modeling through automatic evaluation and analyze case studies to further understand the model's capabilities and limitations. Our code and relevant datasets are publicly available.[1]

## 1 Introduction

Natural language processing (NLP) holds considerable promise in expediting the discovery and understanding of novel material systems, which will be crucial for addressing contemporary societal challenges like climate change and drug discovery. The potential impact of NLP in materials science is chiefly underpinned by the vast reservoir of materials science knowledge contained in text-based resources, such as textbooks, scientific journals, and assorted reports. In spite of the prospective richness of materials science textual data available from diverse sources, a number of challenges continue to significantly hinder the effective digestion and comprehension of relevant materials science textual knowledge (Song et al., 2023; Kononova et al., 2021). Some of the challenges relate to the general availability of data, while other relate to the ability to effectively process domain-specific information, such as chemical notation and data contained in figures and tables (Gupta et al., 2023). This scarcity of readily accessible, high-quality text corpora suitable for efficient language model training has in turn slowed the development of comprehensive language models capable of spanning the extensive conceptual range within the highly interdisciplinary materials science field.

While data availability remains an ongoing challenge in applying modern NLP tools for materials science, recent advancements have led to the emergence of large language models (LLMs) proficient in handling general language tasks that concurrently demonstrate substantial aptitude in areas like chemistry and materials science (Bran et al., 2023; Boiko et al., 2023). Such advancements provide the potential to harness the implicit knowledge encapsulated in these models, which have been trained on vast text corpora spanning a broad range of subjects, to generate accessible, instruction-based datasets for specialized domains like materials science.

Yet, while we can generate targeted instruction-based data to make applying NLP for materials science more accessible, the quality of these instructions requires rigorous evaluation before be-

---

*Equal advising.

†Corresponding author. Canada CIFAR AI Chair.

[1]https://github.com/BangLab-UdeM-Mila/NLP4MatSci-HoneyBee

**MatSci-Instruct Example**

**Instruction:** How was the ME coupling down to the 2D limit verified in the study?

**Input:** [...] In particular, stripy antiferromagnets often exhibit broken crystal and magnetic symmetries, which may bring about the magnetoelectric (ME) effect and enable the manipulation of intriguing properties and functionalities by electrical means. [...] This work reports the ME effect in the 2D stripy antiferromagnetic insulator CrOCl down to a single layer. By measuring the tunneling resistance of CrOCl on the parameter space of temperature, magnetic field, and applied voltage, we verified the ME coupling down to the 2D limit and unraveled its mechanism. [...]

**Output:** The ME coupling down to the 2D limit was verified by measuring the tunneling resistance of CrOCl on the parameter space of temperature, magnetic field, and applied voltage in the study.

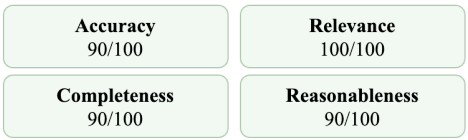

Figure 1: Example instruction generated by the MatSci-Instruct process used to train the HoneyBee language model that contains general language knowledge and is specialized in materials science. The relevant text to correctly answer the instruction is highlighted. MatSci-Instruct follows a structured instruction generation template and ensures instruction quality through an iterative verification loop described in Section 3.1.

ing utilized for language model training. This is particularly salient in the context of complex scientific applications like materials science, which encompasses a wide range of subfields that together describe the properties and behavior of matter that make up materials systems. This need for trustworthy and pertinent instructions necessitates the creation of a robust process to validate the quality of instructions for downstream applications.

Aside from data scarcity in scientific domains, another significant impediment to the application of NLP in materials science is the limited presence of specialized language models that incorporate both in-depth materials science knowledge and a robust understanding of general language. The bulk of today's available language models for materials science are built on the BERT architecture (Gupta et al., 2022; Walker et al., 2021; Huang and Cole, 2022), whose performance in general NLP tasks has been superseded by several more advanced language model architectures in recent years (Touvron et al., 2023; Scao et al., 2022; Brown et al., 2020; Chung et al., 2022). This highlights the need for the development of more capable language models in materials science that can accommodate a

broader knowledge base while effectively performing pertinent materials science language tasks.

This paper seeks to concurrently address the previously outlined challenges of trustworthy instruction generation and capable, open-source language models for materials science. We propose MatSci-Instruct to generate reliable, instruction-based data from large language models. This data is then used to train HoneyBee, a billion-parameter specialized materials science language model based on the LLaMa architecture (Touvron et al., 2023). The key contributions of our research are as follows:

- **MatSci-Instruct – A Two-Step Framework for Trustworthy Instruction-Data Generation:** We propose a universally applicable methodology suited for instruction-data generation in scientific domains. MatSci-Instruct generates specialized instruction-data using a two-step framework – Generation and Verification. In the Generation step, an instructor model (Chat-GPT [2]) creates domain-specific instruction-data focused on materials science. During the Verification step, the instruction-data are cross-verified by a separate verifier model (Claude [3]) for accuracy and relevance as shown by the example in Figure 1. Moreover, in Section 4.1 we conduct human evaluations that suggest good alignment of our generated *MatSci-Instruct Dataset* with human experts across several dimensions: accuracy against known facts, relevance to materials science, and the completeness and reasonableness of the language model output.

- **HoneyBee – A High-Performance LLaMa-Based Model Progressively Trained via MatSci-Instruct:** Utilizing the MatSci-Instruct two-step framework, we apply a Progressive Refinement-Feedback strategy to finetune a LLaMa model, culminating in the HoneyBee model. In our progressive finetuning strategy, the HoneyBee model's performance on MatSci-Instruct data guides subsequent instruction-data generation. This iterative process results in further refined instructions to generate higher-quality instruction-data for finetuning, ensuring the progressive acquisition of specialized knowledge by the

---

[2]https://platform.openai.com/docs/api-reference/chat

[3]https://docs.anthropic.com/claude/docs

model. We evaluate the performance of HoneyBee using a materials science language benchmark (Song et al., 2023), and thoroughly analyze its strengths and limitations.

## 2 Related Work

**Large Language Models** Large Language Models (LLMs) have gained substantial attention from the NLP research and wider technology communities due to their remarkable proficiency in language understanding and generative tasks. Pioneers like GPT-3 (Brown et al., 2020), with its 175 billion parameters, demonstrated the capacity to capture complex linguistic patterns, and subsequent models like Gopher (Rae et al., 2022), GLM (Zeng et al., 2022), PaLM (Chowdhery et al., 2022), BloomZ (Scao et al., 2022), Chincilla (Hoffmann et al., 2022), and OPT (Zhang et al., 2022) continue to drive progress. Commercial models like ChatGPT (OpenAI, 2022) and Claude (Bai et al., 2022) further expand the landscape of performant LLMs. Compared to commercial LLMs, LLaMa (Touvron et al., 2023) stands out for its greater accessibility and good performance, offering an efficient and accessible platform for domain-specific finetuning in various domains, including materials science.

**NLP for Materials Science** NLP applications within materials science are constrained by the dual shortage of openly accessible, high-quality data and high-performing language models. While strides have been made towards enhancing data availability (Song et al., 2023; Olivetti et al., 2020; Kononova et al., 2021; Gao et al., 2020), the primary focus has been on generating expert-annotated data for finetuning BERT-based models, which lack the advanced capabilities of contemporary LLMs. For a detailed review of the performance of various BERT models on materials science language tasks, we refer the reader to Song et al. (2023). The prevailing scarcity of data and specialized LLMs in materials science motivates us to propose MatSci-Instruct, an instruction-based method for data creation, and HoneyBee, a specialized LLM tailored for materials science.

**Instruction Finetuning LLMs** LLMs consistently demonstrate substantial improvements when finetuned for specialized tasks, as seen with biomedical models like ChatDoctor (Li et al., 2023) and HuaTuo (Wang et al., 2023). While the large model size of LLMs poses a challenge for effective finetuning, several efficient methods have been proposed (Mangrulkar et al., 2022), such as P-Tuning (Liu et al., 2021), Prefix Tuning (Li and Liang, 2021), Prompt Tuning (Lester et al., 2021), and LoRA (Hu et al., 2021). Among these, LoRA utilizes low-rank matrix decomposition to limit the additional parameters required for fine-tuning. For data curation in specialized fields, instructions-based fine-tuning extracts detailed data directly from LLMs (Ouyang et al., 2022), reducing human annotation effort and providing scalable solutions. For example, Alpaca (Taori et al., 2023; Wang et al., 2022) exploits LLMs to generate synthetic instructions for model finetuning. However, LLM-synthesized data still suffer from data quality issues, which is especially critical for scientific domains. To address these concerns, we design a generation-verification strategy for trustworthy data generation and a progressive refinement-feedback strategy for finetuning LLMs on specialized instructions.

## 3 Method

Our work consists of two interacting components: 1) *MatSci-Instruct*: a trustworthy instruction generation framework for obtaining scientific textual data from LLMs; 2) *HoneyBee*: a materials science LLM progressively finetuned from LLaMA (Touvron et al., 2023) using MatSci-Instruct generated data. We connect HoneyBee to MatSci-Instruct with a refinement-feedback loop to progressively generate new data and finetune HoneyBee based on its training status as shown in Figure 2.

### 3.1 MatSci-Instruct

The challenges of cost-effectively generating high-quality instruction data are not unique to materials science, but rather, pervasive across various scientific domains. Our proposed solution, *MatSci-Instruct*, is an innovative, domain-agnostic methodology that leverages the power of large language models (LLMs) to generate specialized instruction sets for subsequent model finetuning.

Depicted in Figure 2, *MatSci-Instruct* employs a trifecta of distinct LLMs. The *Instructor* model crafts instructions using structured prompts encapsulating topic and task details. The *Verifier* then evaluates these instructions against accuracy, relevance, completeness, and reasonableness criteria, ensuring only dependable instructions advance to finetuning. Finally, the *Evaluator* assesses the output of the finetuned model along similar dimensions as the Verifier. Poorly executed instructions

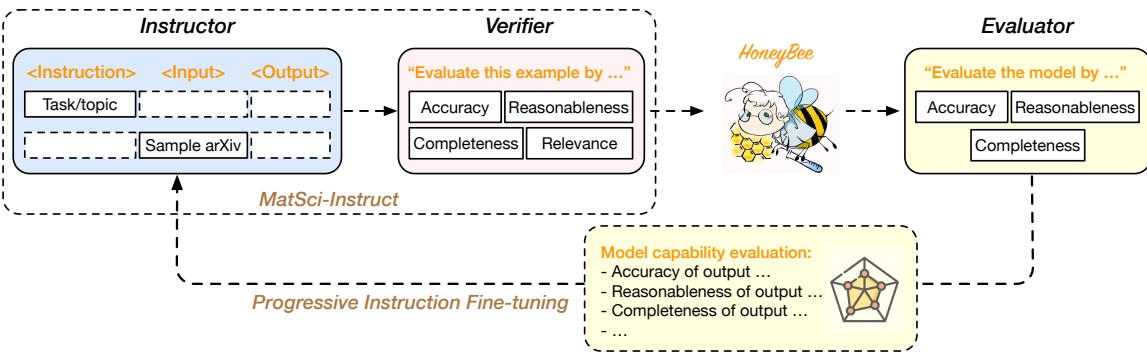

Figure 2: MatSci-Instruct and HoneyBee training workflow. We start with a series of predetermined structured instruction generation prompts that contain both topic and task descriptions. The Instructor (Chat-GPT) then generates a series of instructions that are then passed through the Verfifier (Claude). The instructions that receive high scores with the Verifier are used for progressive finetuning in HoneyBee. The Evaluator (GPT-4) then evaluates HoneyBee's outputs and poor instructions that lead to bad performance are subsequently regenerated from the beginning creating an instruction generation feedback loop for greater instruction quality.

are flagged for further refinement, verification, and evaluation. Ultimately, we generate 52k instructions spanning content-based and open-ended tasks, some of which include empty inputs. Table 1 shows that the number of instructions gets reduced in later stages of the progressive-refinement-feedback loop mainly due to greater emphasis on quality.

| MatSci-Instruct Statistics | |
| --- | --- |
| # instructions for first stage | 52,658 |
| # open-ended instructions | 9,931 |
| # content-based instructions | 39,170 |
| # instructions with empty input | 3,557 |
| # instructions for subsequent stages | 3,020 |
| avg. input length (in words) | 920.8 |
| avg. instruction length (in words) | 76.5 |
| avg. output length (in words) | 211.2 |

Table 1: Statistics of instruction data generated by MatSci-Instruct spanning diverse instruction types.

### 3.1.1 MatSci-Instruct Example

Next, we show a full example of the progressive MatSci-Instruct instruction-data refinement procedure in the loop with HoneyBee model finetuning:

1. Instruction-Data Generation and Finetuning

   - *Data Generation:* Instructor generates training data ($data\_train_0[1-10]$).
   - *Data Verification:* Verifier removes low-scoring data ($data\_train_0[1,2]$)
   - *Finetuning:* LLaMa-7b becomes HoneyBee-7b-Stage-1 after finetuning with data from ($data\_train_0[3-10]$).

2. Evaluation

   - HoneyBee-7b-Stage-1 performs inference on new test data ($data\_test_0$), crafted by the Instructor, with outputs evaluated by the Evaluator.

3. Feedback Response

   - *Response Generation:* HoneyBee-7b-Stage-1 generates responses for the test data ($data\_test_0$).
   - *Scoring Responses:* The Evaluator spots weak responses ($data\_test_0[7,8]$).

4. Instruction-Data Adaptation and Refinement

   - *Focusing on Weaknesses:* The Instructor crafts more training ($data\_train_1$) and test data ($data\_test_1$), focusing on issues identified by the Evaluator.
   - *Finetuning Stage 2:* HoneyBee-7b-Stage-1 refines to HoneyBee-7b-Stage-2.
   - *Re-Evaluation:* HoneyBee-7b-Stage-2 is tested with new test data ($data\_test_1$).

This process is repeated in an iterative feedback loop for continued refinement as shown in Figure 2.

### 3.1.2 Instructor Module

The *Instructor* module of our framework, embodied by ChatGPT, performs the generation of material science instruction-data. This module employs a concise instruction prompt schema composed of three elements: <instruction>, <input>, and <output>. The <instruction> outlines the task using a standardized NLP task set, the <input>

Figure 3: Wordcloud of diverse materials science topics contained in the MatSci-Instruct dataset.

contains the relevant data, and the `<output>` generates a pertinent response to the task.

We query ChatGPT with this schema, populating the `<instruction>` and `<input>` fields with a selection of 20 NLP tasks and 20 materials science subtopics shown in Figure 3, to ensure task and content diversity. These selections are manually verified before they are utilized in structured prompts for generating detailed finetuning instruction-data. Detailed lists of prompts and materials science topics are available in Appendix B and Appendix E.

Following the schema, we engage in a random sampling process, selecting five candidate topics and five tasks, then applying them to the instruction prompts for data generation. For robustness, we direct ChatGPT to flag in the `<output>` field any instruction that cannot be processed based solely on the `<input>` and `<instruction>`. To control task difficulty and boost diversity, we occasionally limit the length of `<instruction>` or `<output>`.

To enhance the diversity and robustness of the instruction-data generation process, our design incorporates several additional strategies. One such strategy employs an open-ended task where the `<input>` field remains intentionally blank, allowing the model to generate responses without pre-defined constraints. This approach tests the generative abilities of the model under uncertainty and promotes more varied outcomes. Another key strategy is content-based instruction-data generation. Instead of relying on predefined topics and tasks, this approach utilizes real-world materials science literature. We select a random open-access paper from the materials science category on arXiv and extract a specific fragment to fill the `<input>` field.

This method not only diversifies the task prompts but also aligns the generated instruction-data more closely with practical, domain-specific contexts.

To conclude the instruction-data generation process, ChatGPT compiles ten representative instruction prompt samples from all possible options. These samples are formatted in a standardized JSON format, readily available for use in the subsequent steps of the *MatSci-Instruct* process. This approach ensures a comprehensive and diverse set of instructions, which in turn contributes to a robust and adaptable language model during finetuning.

### 3.1.3 Verifier Module

Generating high-quality instruction-data can be challenging, and the presence of low-quality data in finetuning a model can lead to misleading results. To address this issue, MatSci-Instruct employs a two-step framework by incorporating a *Verifier* model to improve the trustworthiness of generated data. Specifically, we use Claude as the *Verifier* to ensure the quality of the instructions generated by the *Instructor* (Chat-GPT).

Our evaluation is based on four dimensions: accuracy, relevance, completeness, and reasonableness. Similar to the instruction-data generation, instruction verification is based on a standard set of prompts, shown in Appendix E, which include precise definitions of the evaluation criteria along with the complete instructions generated by the *Instructor*. Concretely, the evaluation criteria are:

- **Accuracy**: The accuracy of the instruction-data is evaluated by comparing it with known facts or credible sources. This involves checking the accuracy of any claims or statements made in the text and verifying that they are supported by evidence.

- **Relevance**: The relevance of the instruction-data is assessed by determining how directly it relates to materials science. This is achieved by analyzing the text's content and ascertaining its applicability to the field.

- **Completeness**: Completeness is an essential dimension to ensure the instruction-data comprehensively address the given task, inclusive of all sub-questions. This involves considering both depth and conciseness to ensure that the output is complete and comprehensive.

- **Reasonableness**: The reasonableness of the instruction-data is about logical consistency.

This dimension ensures no evident contradictions exist within the generated data.

The verifier module (i.e., Claude) evaluates the instruction-data based on the four dimensions mentioned above and identifies any low-quality data that falls below a predetermined threshold. This rigorous verification ensures the use of high-quality data in model fine-tuning, thereby improving the overall efficacy and accuracy of the system. Our verification protocol is designed for modularity and extensibility. This modular design facilitates the incorporation of additional agents into a multi-agent system, each assessing instruction-data based on pre-defined criteria. The final decision on data quality is then reached through a consensus mechanism, augmenting the robustness and comprehensiveness of the verification process, ensuring high-quality data for model finetuning.

### 3.1.4 Evaluator Module

The *Evaluator* model assesses the output of the HoneyBee language model along similar evaluation dimensions as the *Verifier*, namely: accuracy, completeness, and reasonableness. We no longer consider relevance at this stage since the verification step filtered out all instructions with little relevance to materials science. In this paper, we use GPT-4 [4] (OpenAI, 2023) as the *Evaluator* model, which provides an additional independent LLM that is different, and potentially more advanced, than the *Instructor* and *Verifier* LLMs. The *Evaluator* also helps with the identification of poorly formulated instructions according to the performance of the HoneyBee model. These instructions are then passed back to the *Instructor* for additional iterative refinement.

### 3.2 HoneyBee

Upon obtaining a set of trustworthy instruction data from the *Verifier*, we can use the generated instruction dataset to finetune a LLaMa-based model for a specific domain. In this work, we finetune a model for materials science using a progressive finetuning technique to convert a standard LLaMa model to a specialized model in material science: HoneyBee.

### 3.2.1 Progressive Instruction Finetuning

In our approach, as depicted in Figure 2, we harness a progressive instruction finetuning methodology that relies on a feedback loop. This loop enables

---

[4]https://openai.com/research/gpt-4

the progressive generation of new instruction data that takes into account the evaluated model's performance on different criteria, tasks, and topics.

Instructions leading to suboptimal performance by HoneyBee are returned to the *Instructor*, triggering the creation of more detailed and targeted instructions for future iterations. This iterative process also includes instruction evaluation by the *Instructor*, enabling the generation of more precise instruction data for subsequent rounds. For instance, should HoneyBee score low on 'Completeness' for a particular instruction, we inform the *Instructor* of this deficiency, providing the criteria for 'Completeness'. Consequently, the *Instructor* generates enhanced instruction-data to improve HoneyBee's completeness in responding to similar tasks.

Our progressive finetuning process for the language model is based on LoRA (Hu et al., 2021), where we create and train a separate set of low-rank matrices $\psi$ that bypass the need for changing the actual parameters of the language model $\phi$. Since $\psi$ consists of low rank-matrices, it is significantly more parameter and compute efficient for model finetuning. In our finetuning process, we assume that the *Instructor* + *Verifier* models act as the teacher model and the HoneyBee model acts as the student model. In this setting, the student model will continually learn from the instruction-data and undergo testing during the learning process, allowing us to monitor its performance in real-time. The finetuning process continues for a set number of epochs with early stopping if the student model converges to a given loss value. Next, we evaluate the response quality of the student model for any given instruction with the *Evaluator*. In our progressive finetuning strategy, we monitor the evaluation scores after each stage, denoted as $S_{val_{best}}$, and terminate the process when the $S_{val_{best}}$ stops yielding significant improvements. In our experiments in Section 4, we perform three stages of progressively finetuning both the instruction-data and the HoneyBee model parameters.

## 4 Experiments

Our experiments focus on assessing the ability of MatSci-Instruct to create high-quality, trustworthy instruction-data relevant to materials science, as described in Section 3.1, along with understanding the capabilities and limitations of HoneyBee.

## 4.1 MatSci-Instruct Evaluation

A critical piece of the MatSci-Instruct pipeline is the independent verification and evaluation of the instruction-data generated by the Instructor model. Given the importance of the Verifier and Evaluator in ensuring the quality of the instruction-data, and the fact that understanding materials science textual data requires deep domain understanding, we conducted an evaluation with human experts on the trustworthiness of the instructions generated by MatSci-Instruct. In our human expert evaluation, we asked two graduate students majoring in material science to evaluate 50 randomly selected instruction data along the same evaluation dimensions as the Verifier module (accuracy, relevance, completeness, reasonableness). Next, we conducted a verification and evaluation of the same 50 instructions using Claude and GPT-4 respectively. We measure agreement between the human experts and the LLMs by calculating Spearman and Pearson correlation coefficients between the scores along each of the dimensions.

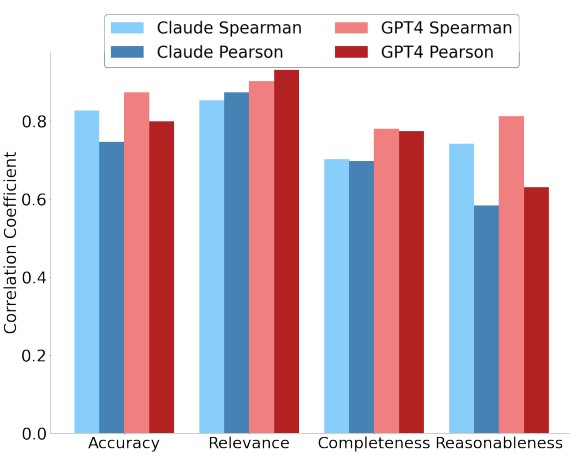

Figure 4: Correlation between human evaluation and LLM evaluation (Claude, GPT-4). Both Spearman and Pearson correlation coefficients consistently exceed 0.6 between both methods indicating good agreement.

As shown in Figure 4, both Claude and GPT-4 had correlation coefficients higher than 0.6 for each dimension and an overall coefficient as high as 0.8 when compared to manual evaluation. This indicates a decent level of consistency between manual and automatic evaluations for a random sample of instructions, which gives us confidence in the ability of MatSci-Instruct to generate trustworthy, high-quality instructions for HoneyBee finetuning.

| Model | Accuracy | Completeness | Reasonableness |
|---|---|---|---|
| **Zero-Shot LLMs** | | | |
| Chat-GPT | 92.55 | 98.74 | 99.84 |
| Llama-7b | 78.81 | 90.36 | 97.64 |
| Llama-13b | 84.22 | 91.22 | 98.33 |
| Alpaca-7b | 81.35 | 92.01 | 98.49 |
| Alpaca-13b | 86.24 | 92.17 | 98.80 |
| **HoneyBee without Verfication** | | | |
| HoneyBee-7b | 85.42 | 93.24 | 98.49 |
| HoneyBee-13b | 88.76 | 93.99 | 98.93 |
| **HoneyBee with MatSci-Instruct** | | | |
| HB-7b-Stage1 | 88.81 | 93.42 | 99.07 |
| HB-7b-Stage2 | 89.99 | 94.84 | 99.64 |
| HB-7b-Stage3 | 91.95 | 95.78 | 99.90 |
| HB-13b-Stage1 | 94.17 | 94.42 | 99.40 |
| HB-13b-Stage2 | 96.42 | 95.42 | 99.78 |
| HB-13b-Stage3 | 98.11 | 97.00 | 99.89 |

Table 2: Evaluation results for various LLMs based on performance on MatSci-Instruct data along with accuracy, completeness, and reasonableness performed by GPT-4. HoneyBee performs better with verification and gets progressively better with each iterative stage of MatSci-Instruct approaching and exceeding the performance of Chat-GPT in the case of HoneyBee-13b. We highlight scores that outperform Chat-GPT.

## 4.2 HoneyBee Task Evaluation

We generated a test set of instruction-data with the same generation process as the train set that is only introduced it during model evaluation. The results in Table 2 show that HoneyBee gets progressively better with each iteration of MatSci-Instruct for both HoneyBee-7b and HoneyBee-13b. HoneyBee without verification also outperforms LLaMA and Alpaca LLMs of equal size indicating the value of the progressive finetuning approach on specialized materials science instruction-data. HoneyBee-13b closely matches, and in some exceeds, the evaluation performance of Chat-GPT which served as the Instructor. Notably, HoneyBee-13b is $\sim 10x$ more parameter efficient than GPT-3.

## 4.3 HoneyBee Performance on MatSci-NLP

In addition to evaluating the performance of HoneyBee based on LLM assessment in Section 4.2, we investigate the performance of HoneyBee on MatSci-NLP, a broad benchmark of materials science NLP tasks (Song et al., 2023). We study HoneyBee's performance under two settings: 1. Low-data training setting as applied in the original paper by Song et al. (2023); 2. Zero-shot performance

Table 3: Low-resource finetuning and zero-shot evaluation results for various HoneyBee on MatSci-NLP tasks. For low-resource finetuning, we follow the method described in Song et al. (2023). HoneyBee outperforms all models across the vast majority of tasks for both low-resource finetuning and zero-shot settings. MatSci-Instruct's Progressive-Refinement-Feedback method improves HoneyBee's performance for each consecutive stage. We report macro-F1 (top) and micro-F1 (bottom) scores highlighting the best, second-best and third-best performing LLM. Honey-7b and HoneyBee-13b outperform both ChatGPT and Claude and are generally competitive with GPT-4.

| Model | Named Entity Recognition | Relation Extraction | Event Argument Extraction | Paragraph Classification | Synthesis Action Retrieval | Sentence Classification | Slot Filling | Overall (All Tasks) |
|---|---|---|---|---|---|---|---|---|
| **Low-Resource Finetuning on MatSci-NLP** | | | | | | | | |
| MatSciBERT | 0.707 | 0.791 | 0.436 | 0.719 | 0.692 | 0.914 | 0.436 | 0.671 |
| (Gupta et al., 2022) | 0.470 | 0.507 | 0.251 | 0.623 | 0.484 | 0.660 | 0.194 | 0.456 |
| MatBERT | 0.875 | 0.804 | 0.451 | 0.756 | 0.717 | 0.909 | 0.548 | 0.722 |
| (Walker et al., 2021) | 0.630 | 0.513 | 0.288 | 0.691 | 0.594 | 0.614 | 0.273 | 0.517 |
| HoneyBee-7b | 0.787 | 0.852 | 0.551 | 0.741 | 0.792 | 0.991 | 0.529 | 0.749 |
| | 0.644 | 0.518 | 0.389 | 0.641 | 0.617 | 0.711 | 0.391 | 0.559 |
| HoneyBee-13b | 0.860 | 0.921 | 0.653 | 0.761 | 0.853 | 0.998 | 0.554 | 0.80 |
| | 0.748 | 0.578 | 0.486 | 0.658 | 0.662 | 0.743 | 0.401 | 0.611 |
| **Zero-Shot LLM Performance** | | | | | | | | |
| LLaMA-7b | 0.042 | 0.094 | 0.160 | 0.279 | 0.052 | 0.096 | 0.142 | 0.208 |
| (Touvron et al., 2023) | 0.064 | 0.013 | 0.042 | 0.218 | 0.013 | 0.087 | 0.010 | 0.064 |
| LLaMA-13b | 0.057 | 0.109 | 0.042 | 0.233 | 0.039 | 0.079 | 0.138 | 0.1 |
| (Touvron et al., 2023) | 0.066 | 0.016 | 0.054 | 0.189 | 0.009 | 0.074 | 0.008 | 0.059 |
| Alpaca-7b | 0.031 | 0.053 | 0.029 | 0.375 | 0.179 | 0.180 | 0.139 | 0.141 |
| (Taori et al., 2023) | 0.018 | 0.037 | 0.009 | 0.294 | 0.129 | 0.180 | 0.039 | 0.101 |
| Alpaca-13b | 0.053 | 0.016 | 0.111 | 0.310 | 0.442 | 0.375 | 0.110 | 0.202 |
| (Taori et al., 2023) | 0.046 | 0.035 | 0.072 | 0.237 | 0.278 | 0.334 | 0.015 | 0.145 |
| Chat-GPT | 0.063 | 0.232 | 0.204 | 0.433 | 0.300 | 0.320 | 0.368 | 0.274 |
| (OpenAI, 2022) | 0.052 | 0.145 | 0.203 | 0.450 | 0.183 | 0.318 | 0.280 | 0.233 |
| Claude | 0.063 | 0.232 | 0.195 | 0.442 | 0.280 | 0.329 | 0.393 | 0.276 |
| (Bai et al., 2022) | 0.048 | 0.143 | 0.169 | 0.467 | 0.177 | 0.326 | 0.305 | 0.234 |
| GPT-4 | 0.189 | 0.445 | 0.453 | 0.679 | 0.743 | 0.788 | 0.502 | 0.543 |
| (OpenAI, 2023) | 0.121 | 0.432 | 0.353 | 0.522 | 0.677 | 0.689 | 0.483 | 0.468 |
| **Zero-Shot HoneyBee with MatSci-Instruct** | | | | | | | | |
| HoneyBee-7b-Stage1 | 0.173 | 0.138 | 0.196 | 0.380 | 0.592 | 0.416 | 0.292 | 0.301 |
| | 0.148 | 0.120 | 0.096 | 0.207 | 0.208 | 0.334 | 0.105 | 0.174 |
| HoneyBee-7b-Stage2 | 0.243 | 0.199 | 0.237 | 0.440 | 0.612 | 0.467 | 0.344 | 0.363 |
| | 0.166 | 0.145 | 0.123 | 0.301 | 0.289 | 0.345 | 0.176 | 0.221 |
| HoneyBee-7b-Stage3 | 0.267 | 0.245 | 0.290 | 0.490 | 0.688 | 0.490 | 0.393 | 0.409 |
| | 0.190 | 0.178 | 0.189 | 0.343 | 0.342 | 0.365 | 0.289 | 0.271 |
| HoneyBee-13b-Stage1 | 0.369 | 0.301 | 0.389 | 0.500 | 0.701 | 0.512 | 0.467 | 0.463 |
| | 0.256 | 0.224 | 0.265 | 0.379 | 0.378 | 0.402 | 0.334 | 0.320 |
| HoneyBee-13b-Stage2 | 0.391 | 0.367 | 0.437 | 0.576 | 0.765 | 0.557 | 0.508 | 0.514 |
| | 0.299 | 0.290 | 0.303 | 0.411 | 0.401 | 0.461 | 0.379 | 0.363 |
| HoneyBee-13b-Stage3 | 0.429 | 0.412 | 0.481 | 0.611 | 0.801 | 0.589 | 0.578 | 0.557 |
| | 0.372 | 0.346 | 0.378 | 0.467 | 0.429 | 0.503 | 0.423 | 0.417 |

on MatSci-NLP tasks shown in Table 3. MatSci-NLP contains a wide range of text data related to material science that spans a wide range of NLP tasks and types of materials, including but not limited to fuel cells, inorganic materials, glasses, and superconductors. For evaluation on MatSci-NLP, we follow the same convention as in Song et al. (2023) where we report both macro-F1 and micro-F1 scores in Table 3.

**Low-Resource Finetuning:** In our experiments, we follow the same low-resource procedure described in (Song et al., 2023) by splitting the data in MatSci-NLP into 1% training subset and a 99% test subset for evaluation. Subsequently, all models, including HoneyBee, are finetuned on the training subset and evaluated on the test subset of MatSci-NLP data. The results on low-resource finetuning in Table 3 show that both HoneyBee-7b and HoneyBee-13b perform best overall, outperform-

ing MatBERT (Walker et al., 2021) and MatSciB-ERT (Gupta et al., 2022) among all tasks in MatSci-NLP with the exception of named entity recognition. MatBERT and MatSci-BERT are both BERT models pretrained on different corpora of materials science textual data. While the domain-specific pretraining significantly boosts the score of both models for MatSci-NLP tasks, HoneyBee shows better performance without requiring pretraining on materials science textual data. This is a significant advantage of HoneyBee and MatSci-Instruct given that large, high-quality corpora of materials science text are generally difficult to obtain as described in Section 2.

**Zero-Shot Performance:** The zero-shot performance results, where we assess performance directly on the test subset, in the lower part of Table 3 show that HoneyBee outperforms both LLaMa and Alpaca models. Notably, HoneyBee-7b-Stage1, which corresponds to only one round of MatSci-Instruct, outperforms both LLaMa and Alpaca models for equal (7b) and larger (13b) parameter sizes. The data in Table 3 further confirms the results from Table 2 that show progressive improvement with each stage of MatSci-Instruct where both HoneyBee-7 and Honey13b exhibit clear improvement in iterative stages. We also observe that model parameter size matters for zero-shot performance with 13b parameter models outperforming 7b for HoneyBee and Alpaca, both of which are instruction finetuned models. Interestingly, LLaMA-7b generally outperforms LLaMa-13b across most MatSci-NLP tasks and in the overall score on MatSci-NLP.

### 4.4 HoneyBee — Case Study

We perform a case study to further understand the capabilities and limitations of the various LLMs we studied, including HoneyBee, Alpaca, and Chat-GPT. Our case study results, with full data and text included in Appendix D, show that HoneyBee-13b generally produces outputs of the same quality as Chat-GPT while other models generally produce lower quality outputs. This provides additional weight to the results in Section 4.1 indicating that HoneyBee-13b can match the quality of Chat-GPT after multiple rounds of progressive refinement-feedback finetuning using MatSci-Instruct.

## 5   Conclusion

In this work, we introduce MatSci-Instruct, an iterative instruction generation method for materials science, and HoneyBee, a state-of-the-art large language model for materials science. To the best of our knowledge, HoneyBee is the first billion-parameter scale language model that is specialized in materials science. HoneyBee outperforms current state-of-the-art general language models (LLaMa, Alpaca) and materials science BERT-based language models (MatBERT, MatSciBERT) in various materials science NLP tasks, and note HoneyBee's performance improvement with each successive MatSci-Instruct stage. MatSci-Instruct provides a valuable framework for generating instruction-data to progressively finetune LLMs where instruction-data from an Instructor are verified by a Verifier before being used for finetuning. Additionally, poor instruction-data is refined based on feedback from an Evaluator leading to higher quality instruction-data and model performance for the desired specialization as shown by the results in Section 4. Future work remains in augmenting materials science LLMs with external knowledge, such as known scientific facts, which can further improve an LLM's reliability and interpretability.

## Limitations

While HoneyBee outperforms current state-of-the-art methods in various materials science NLP tasks, it remains unclear how well HoneyBee would generalize the tasks outside of the MatSci-NLP benchmark and MatSci-Instruct instruction-data to solve complex materials science challenges. Such challenges may include creating a synthesis recipe for a new materials or explaining the behavior of a materials system based on fundamental scientific concepts. Materials science remains a wide-ranging and complex field with many open questions remaining on the true capability of HoneyBee and other LLMs to understand important materials science concepts. MatSci-Instruct also relies on the availability of highly performant LLMs to serve as the Instructor, Verifier and Evaluator which can be limited in their own capabilities. Furthermore, our work focuses primarily on the materials science domain and further studies are needed to understand how applicable our methods would be to additional scientific domains.

## Broader Impact

Both HoneyBee and MatSci-Instruct can help promote research on NLP for material science in applying, training and evaluating LLMs for practical applications in the field. The general frameworks described in this paper can also be transferred to other scientific domain, such biology, physics and chemistry, where trustworthy textual data is required.

Our research does not raise major ethical concerns.

## Acknowlegments

This work is supported by the Mila internal funding - Program P2-V1: Industry Sponsored Academic Labs (project number: 10379), the Canada CIFAR AI Chair Program, and the Canada NSERC Discovery Grant (RGPIN-2021-03115).

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

# Appendix

## A  Experimental Details

We finetune LLaMA models with 7B and 13B parameters using instructions from MatSci-Instruct. As shown in Table 2, we analyze the effect of various rounds of iterative instruction feedback. For finetuning, we use the AdamW optimizer with an initial learning rate of 1e-4 on 2 A100 GPUs for LLaMa-7b and 4 A100 GPUs for LLaMa-13b. We assign a batch size of 4 to each GPU with a gradient accumulation step of 32 and a maximum sequence length of 2048.

## B  Instruction Generation Details

Table 4: MatSci-Instruct samples a diverse set of materials science topic areas.

| MatSci-Instruct Topics | |
|---|---|
| Bio-inspired Materials | 221 |
| Self-Healing Materials | 209 |
| Magnetic Materials | 195 |
| Smart Materials | 190 |
| Metals | 189 |
| Semiconductors | 188 |
| Carbon Nanotubes | 184 |
| Polymers | 182 |
| Thermoelectric Materials | 180 |
| Optical Materials | 180 |
| Superconductors | 179 |
| Graphene | 177 |
| Glass | 174 |
| Energy Materials | 166 |
| Composites | 165 |
| Electronic Materials | 163 |
| Construction Materials | 158 |
| Ceramics | 155 |
| Nanoaterials | 153 |
| biomaterials | 149 |

The full list of materials science topics and NLP tasks sampled for MatSci-Instruct instruction are included in Table 4 and Table 5. We sample a broad range of materials science topics and NLP that are generally balanced yielding a set of instructions that includes specialized materials science text as well as general language capabilities.

Table 5: MatSci-Instruct samples a diverse set of NLP tasks to generate instructions including general NLP tasks to main general language capabilities.

| MatSci-Instruct NLP Tasks | |
|---|---|
| Machine Reading Comprehension | 224 |
| Question Answering | 224 |
| Open-Ended Generation | 214 |
| Classification | 203 |
| Information Extraction | 201 |
| Relation Extraction | 193 |
| Analysis | 189 |
| Topic Modeling | 188 |
| Writing | 180 |
| Commonsense Reasoning | 172 |
| Code Interpretation | 172 |
| Event Extraction | 167 |
| Grammar Correction | 165 |
| Clustertin | 162 |
| Named Entity Recognition | 160 |
| Text Simplification | 153 |
| Summarization | 149 |
| Sentiment Analysis | 149 |
| Part-of-Speech Tagging | 146 |
| Editing | 145 |

## C  Quality of Instructions for each MatSci-Instruct Stage

The preset threshold for filtering low-quality data is an average score of 95 or above for all four dimensions and a score of no less than 90 for each individual dimension. As shown in Table 6, after filtering, the remaining instruction data exhibits significantly improved quality, approaching full marks making the data significantly more trustworthy for finetuning specialized LLMs.

## D  LLM Case Study

This section provides the details for the case study in Section 4.4.

### D.1  Open-ended Instruction

We analyze the response of different LLMs to the following open-ended query: *Analyze the properties and applications of electronic materials.*

- **Chat-GPT:** Electronic materials are a class of materials that possess unique properties that make them suitable for use in electronic devices and technologies. These materials are characterized by their electrical conductivity,

Table 6: Claude evaluation scores of MatSci-Instruct before and after removing low-quality instruction data

| Score/Stage | | 1 | 2 | 3 |
|---|---|---|---|---|
| avg. Accuracy | before | 92.45 | 92.20 | 92.56 |
| | after | 100.0 | 100.0 | 100.0 |
| avg. Relevance | before | 86.01 | 87.18 | 86.96 |
| | after | 100.0 | 100.0 | 100.0 |
| avg. Completeness | before | 88.80 | 86.13 | 89.33 |
| | after | 99.98 | 99.99 | 99.99 |
| avg. Reasonableness | before | 97.75 | 98.21 | 98.16 |
| | after | 100.0 | 100.0 | 100.0 |

optical properties, magnetic properties, and thermal properties. The electrical conductivity of electronic materials is a critical property for materials used in electronic devices such as transistors, diodes, and integrated circuits. Semiconductors such as silicon and germanium are widely used in microprocessors and solar cells because of their unique electrical properties. Optical properties are also important for electronic materials used in displays, touch screens, and flat panel displays. Optoelectronic materials like indium tin oxide (ITO) and gallium nitride (GaN) possess unique optical properties that make them ideal for use in these devices. Magnetic properties are important for electronic materials used in magnetic sensors, hard drives, and magnetic memory. Magnetic materials like iron, cobalt, and nickel possess unique magnetic properties that make them ideal for use in these devices. Lastly, thermal properties are important for electronic materials used in high-temperature electronic devices such as microprocessors and power electronics. Materials like diamond and silicon carbide possess unique thermal properties that make them ideal for use in these devices.

- **HoneyBee-13b:** Electronic materials have unique properties such as electrical conductivity, optical properties, magnetic properties, and thermal properties, making them suitable for use in electronic devices. Semiconductors such as silicon and germanium are used in microprocessors and solar cells due to their unique electrical properties. Optoelectronic materials like indium tin oxide (ITO) and gal-

lium nitride (GaN) possess unique optical properties, ideal for use in displays, touch screens, and flat panel displays. Magnetic materials like iron, cobalt, and nickel are used in magnetic sensors, hard drives, and magnetic memory due to their unique magnetic properties. Materials like diamond and silicon carbide possess unique thermal properties ideal for use in high-temperature electronic devices. The properties of electronic materials are critical for modern electronic devices, and research is ongoing to discover new materials with even more unique properties.

- **HoneyBee-7B:** Electronic materials possess unique properties that make them suitable for various applications. They exhibit high electrical conductivity, low resistivity, and excellent thermal conductivity. These materials are commonly used in electronic devices such as transistors, integrated circuits, and sensors. They also find applications in optoelectronics, energy storage devices, and flexible electronics. The properties of electronic materials enable efficient charge transport and facilitate the flow of electrons, making them essential components in modern technology.

- **Alpaca-13b:** Electronic materials are materials that have electrical properties and can be used in a variety of applications. These materials can be classified into two categories: conductors and insulators. Conductors are materials that allow the flow of electricity, while insulators are materials that do not allow the flow of electricity. Electronic materials can be used in a variety of applications, such as in semiconductors, transistors, and diodes. Semiconductors are materials that have properties that are intermediate between those of conductors and insulators and are used in the manufacture of integrated circuits. Electronic materials are also used in the manufacture of solar cells, which convert light into electricity.

- **Alpaca-7b:** Electronic materials are materials that have the ability to control the flow of electrons. These materials are used in a variety of applications, such as electronics, optics, and sensors. Examples of electronic materials include semiconductors, which are materials that can be doped to control the flow of electrons, and metallic materials, which are con-

ductors of electricity. Other examples include superconductors, which are materials that can conduct electricity with zero resistance, and dielectrics, which are materials that can store electricity. Electronic materials are used in a variety of applications, such as electronics, optics, and sensors.

ChatGPT is the most comprehensive and relevant answer as it provides a detailed analysis of the properties and applications of electronic materials, covering electrical conductivity, optical properties, magnetic properties, and thermal properties. Chat-GPT also gives specific examples of materials and their applications in various electronic devices. HoneyBee-13b is the second-best answer as it also covers the key properties of electronic materials and their applications in various electronic devices. HoneyBee-13b provides specific examples of materials and their unique properties that make them ideal for use in different applications. However, HoneyBee-13b's output lacks specificity and does not provide enough detail to fully address the question. HoneyBee-7b gives the third-best answer as it gives a general overview of the unique properties of electronic materials and their applications in various electronic devices. However, HoneyBee-7b's output lacks specificity and does not provide enough detail. Alpaca-13b is the fourth-best answer as it provides a basic understanding of electronic materials and their use in various applications but lacks detail and specificity. Alpaca-7b is the least comprehensive and relevant answer as it provides a limited understanding of electronic materials and their use in various applications. Alpaca-7b's output does not provide enough detail or specific examples to support its claims.

### D.2 Content-based Instruction

Next, we analyze the response of different LLMs to the following query: *What are the exciton interactions responsible for?* with the following <input>: " Inspired by the recent experimental observation of strongly coupled polaritons in a Moire heter-obilayer, we study a model of dipole-interacting excitons localized on sites of a lattice and coupled to planar cavity photons. We calculate the phase diagram of this system by computing fluctuations around the mean field and determining the stability of the excitation spectrum. We find that the transition from the normal state to a polariton condensate is intermediated by a series of ordered states at partial fillings of the exciton lattice, stabilized by the exciton interactions. In particular we predict a supersolid phase in which a polariton condensate coexists with spatial order. " The replies from the different LLMs are below:

- **Chat-GPT:** The exciton interactions are responsible for stabilizing the ordered states at partial fillings of the exciton lattice, and in particular for predicting a supersolid phase in which a polariton condensate coexists with spatial order.

- **HoneyBee-13b:** The exciton interactions are responsible for stabilizing the ordered states at partial fillings of the exciton lattice, which are intermediates between the normal state and a polariton condensate. In particular, they are responsible for the predicted supersolid phase, where a polariton condensate coexists with spatial order.

- **HoneyBee-7b:** The exciton interactions are responsible for stabilizing the ordered phases that mediate the transition to the polariton condensate, including the supersolid phase.

- **Alpaca-13b:** The exciton interactions studied in the given text are responsible for the phase transition from the normal state to a polariton condensate, as well as the supersolid phase

- **Alpaca-7b:** The exciton interactions are responsible for the transition from the normal state to a polariton condensate, as well as the stabilization of the supersolid phase

The answers from HoneyBee-13b and Chat-GPT are better compared to the outputs of the other models. HoneyBee-13b and Chat-GPT directly address the question by mentioning the ordered states at partial fillings of the exciton lattice, which are intermediates between the normal state and a polariton condensate, and the predicted supersolid phase. The answers also use language that closely matches the language used in the original text, indicating a good understanding of the material.

### E LLM Prompts

In this section we provide some of the prompts used for the different modules in MatSci-Instruct. We plan to make the full list of prompts, data and code available upon publication.

- "Evaluate accuracy of the given text by comparing with known facts or credible sources. This involves checking the accuracy of any claims or statements made in the text, and verifying that they are supported by evidence. The next line directly provide the text. {output_text} Please return a score ranging from 0 to 100, with 0 being the worst and 100 being the best. Please use the strictest grading standard. The score should be in JSON format with a field name of 'score'. You should not output any other information or text."

- "Evaluate relevance of the given text by considering how directly the text is related to materials science. The next line directly provide the text. {output_text} Please return a score ranging from 0 to 100, with 0 being the worst and 100 being the best. Please use the strictest grading standard. The score should be in JSON format with a field name of 'score'. You should not output any other information or text."

- "Evaluate completeness of the given text (including input, instruction and output) by assessing how fully the output addresses the instruction, including all sub-questions. Consider both depth and conciseness. The next 3 lines directly provide the input, instruction and output respectively. {input_text} {instruction} {output_text} Please return a score ranging from 0 to 100, with 0 being the worst and 100 being the best. Please use the strictest grading standard. The score should be in JSON format with a field name of 'score'. You should not output any other information or text."

- "Evaluate reasonableness of the given text by considering how logically consistent the content is, with no obvious contradictions. The next line directly provide text. {output_text} The score should range from 0 to 100, with 0 being the worst and 100 being the best. Please use the strictest grading standard. The score should be in JSON format with a field name of 'score'. You should not output any other information or text."