# OpenReview forum: "HoneyBee: Progressive Instruction Finetuning of Large Language Models for Materials Science"
_EMNLP/2023/Conference — EMNLP 2023 Findings_

### Official Review · Reviewer_ztYe · 2023-08-04

**Soundness:** 3

**Excitement:**

3: Ambivalent: It has merits (e.g., it reports state-of-the-art results, the idea is nice), but there are key weaknesses (e.g., it describes incremental work), and it can significantly benefit from another round of revision. However, I won't object to accepting it if my co-reviewers champion it.

**Paper Topic And Main Contributions:**

This paper proposes an iterative instruction generation method for materials science that can also apply to other scientific domains. In a generation step, ChatGPT generates domain-specific instructions which are then evaluated by an independent evaluator. Additionally, an evaluator refines poor quality instructions based on feedback, leading to higher quality instructions and model performance in the targeted domain. Following this approach, they obtain a billion-parameter LLM specialized in the materials science domain. Experiments on benchmarks demonstrate the superiority of proposed model over existing language models in materials science.

**Reasons To Accept:**

1. Two-step framework to generate trustworthy instructions in target scientific domains. Paper shows effectiveness in material science domain.
2. The billion-parameter specialized materials science LM is a valuable resource, especially since it outperforms existing language models in materials science.
3. The iterative approach shows that finetuning-evaluation feedback can progressively improve the quality of a domain-specialized model.
4. Experiments indicate superiority of the proposed model over existing models in materials science.


**Reasons To Reject:**

1. The paper does not motivate the proposed approach and justify the design choices made.
2. There are many approaches to distill knowledge from large language models (e.g. https://arxiv.org/pdf/2110.07178.pdf) for targeted domains. These approaches are not discussed or compared against. Are these approaches ineffective for scientific domains, and why?
3. The paper does not provide a detailed analysis of the computational resources required for training and fine-tuning.
4. In scientific domains such as material science that are evolving and demand factuality, it might make sense to augment language models with external knowledge (i.e., rely on semi-parametric knowledge) instead of trying to pack everything in the model parameters (parametric knowledge).

**Reproducibility:**

3: Could reproduce the results with some difficulty. The settings of parameters are underspecified or subjectively determined; the training/evaluation data are not widely available.

**Reviewer Confidence:**

4: Quite sure. I tried to check the important points carefully. It's unlikely, though conceivable, that I missed something that should affect my ratings.

---

> ### Author Rebuttal · Authors · 2023-08-29
>
> We thank you for feedback and are happy that you recognize the value of our contribution in creating a trustworthy instruction generation process for finetuning LLMs on materials science. We also appreciate that you recognize that HoneyBee will be a valuable contribution to the community and outperforms current methods on materials science NLP tasks, especially with iterative instruction generation. Next we address some of the feedback and questions you called out:
>
> > Q: The paper does not motivate the proposed approach and justify the design choices made.
>
> Thank you for this feedback. Our primary motivation is to create a high-performing, open-source LLM for materials science (HoneyBee), which does not exist today. In order to create HoneyBee, we required trustworthy text-based data that is relevant to materials science which is not available in the public literature. To generate that data, we developed MatSci-Instruct, which creates instructions-data for finetuning a LlaMa model to arrive at HoneyBee in iterative refinement stages. We chose LlaMa as the base model given its strong performance, open-source availability and past work indicating good performance for instructions based finetuning (e.g., Alpaca). MatSci-Instruct leverages various LLMs as instructor, verifier and evaluator to minimize bias and increase trustworthiness in the instruction-data generation process with iterative refinement leading to additional performance increases. To the best of our knowledge, HoneyBee is the best performing open-source language model in materials science, even outperforming closed source LLMs, which we plan to release to the community at no cost, making it more accessible than closed-source LLMs.
>
> > Q: There are many approaches to distill knowledge from large language models (e.g. https://arxiv.org/pdf/2110.07178.pdf) for targeted domains. These approaches are not discussed or compared against. Are these approaches ineffective for scientific domains, and why?
>
> Compared with instruction finetuning, traditional methods for knowledge distillation in specific fields (such as material science) have the following drawbacks:
> *  **Poor Generalization Ability:** Knowledge distillation provides poor generalization compared to instruction-based finetuning, which unifies all tasks into the form of QA, and can cover various tasks and enhance the model's generalization ability.
> * **Instructions-Data Quality:** In our approach, we can control the instructions of fine-tuning, not only to ensure the quality of the data, but also to generate data purposefully, so as to obtain a better knowledge distillation effect.
> * **In-Context Learning:** Instruction-based fine-tuning can fully explore the in-context learning ability of large models and improve the modeling ability of language models to NLP task instructions. Supervised training can teach language models to learn to complete tasks according to instructions, so that language models can learn to follow instructions, even for tasks they have never seen before.
>
>
> > Q: The paper does not provide a detailed analysis of the computational resources required for training and fine-tuning.
>
> In Appendix A, we outline details for the compute requirements for our experiments: Specifically, we use 2 A100 GPUs for LLaMa-7b experiments and 4 A100 GPUs for LLaMa-13b experiments. Please let us know if you would like to see any additional details.
>
> > Q: In scientific domains such as material science that are evolving and demand factuality, it might make sense to augment language models with external knowledge (i.e., rely on semi-parametric knowledge) instead of trying to pack everything in the model parameters (parametric knowledge).
>
> Thank you for making this very relevant point. In a general sense, by relying on semi-parametric knowledge, language models can benefit from the scalability and flexibility of external knowledge sources, which can be updated independently and adapted to specific application domains. Yet, obtaining external knowledge and modules specifically tailored to the material science domain can be challenging. Material science is a specialized field with complex and domain-specific information that may not be readily available in structured or easily accessible formats. It therefore requires a diverse set of tools for high-quality data curation, as well as domain-specific methods to effectively augment language models with external material science knowledge or modules.
> Given the variety of technical challenges required to properly infuse semi-parametric knowledge, we leave this research for future work. In this case, one can consider augmenting large language models with external knowledge instead of attempting to include all knowledge within the model parameters.

---

### Official Review · Reviewer_RMWP · 2023-08-09

**Soundness:** 3

**Excitement:**

3: Ambivalent: It has merits (e.g., it reports state-of-the-art results, the idea is nice), but there are key weaknesses (e.g., it describes incremental work), and it can significantly benefit from another round of revision. However, I won't object to accepting it if my co-reviewers champion it.

**Missing References:**

None.

**Paper Topic And Main Contributions:**

The paper presents an approach for generating training data using a variety of commercial LLMs (ChatGPT, Claude, GPT4) for fine-tuning billion parameter LLMs (based on LLaMa) for a variety of materials science NLP tasks - the generated data (and the process) is referred to as MatSci-Instruct and the finetuned LLaMa model is referred to as HoneyBee. The training data is generated by ChatGPT, verified by Claude, used to finetune a LLaMa model, the outputs of which are evaluated by GPT4. This process is repeated a few times. The paper evaluates the HoneyBee models on a benchmark of NLP tasks in materials science and demonstrates superior performance to prior methods based on small parameter BERT models.

**Questions For The Authors:**

A. Sec 3.1.1: Do I understand correctly that the generated data from ChatGPT is the **output** for an "instruction" and an "input" rather than the instruction itself? Please clarify and consistently specify the data generated in this work. This section directly conflicts with the abstract: "Moreover, we iteratively **generate more targeted instructions** in a finetuning-evaluation-feedback loop leading to progressively better performance for our finetuned HoneyBee models."

B. "We query ChatGPT with this schema, populating the <instruction> and <input> fields with a selection of 20 NLP tasks and 20 materials science subtopics shown in Figure 3, to ensure task and content diversity." - Please describe the following in greater detail: 1) I see that Table 5 lists the 20 tasks, where were these tasks obtained? i.e is it an existing dataset? 2) What do the second columns of Table 4 and 5 mean? 3) Where was the input fed to ChatGPT for each of the tasks obtained? 4) Was a single instruction per task used for the 20 tasks? 5) Were the instructions manually authored?

C. Was there an overlap in the input data to ChatGPT and the MatSci-NLP tasks?

D. Did ChatGPT, Claude, or GPT4 use any few shot examples in their prompts?

E. Sec 3.1.2: "The accuracy of the instruction data is evaluated by comparing it with known facts or credible sources" - where are the facts and credible sources obtained from?

F. Sec 3.2.1: "Instructions leading to suboptimal performance by HoneyBee are returned to the Instructor, triggering the creation of more detailed and targeted instructions for future iterations." - Setting aside the confusion of question A, if the instruction is flawed and the instruction is changed to fit a poorly trained model is the instruction not likely to be contrived and unintuitive to humans?

G. Sec 4.3: "Lowdata training setting as applied in the original paper by Song et al. (2023)" - please describe in the body of this paper the precise fine-tuning data that was used from Song et al.? Were Honeybee models fine-tuned on any additional data besides the synthetic data generated in MatSci-Instruct?

H. Please consider reporting the performance of ChatGPT on the benchmark dataset of Table 3 (the metrics of Table 2 are vague), this is likely to be extremely useful to practitioners choosing to use Honeybee models vs ChatGPT. This is also likely to be further vet the effectiveness of your data generation process.

**Reasons To Accept:**

- The paper is easy to read.
- The paper reports strong empirical gains over prior baselines.

**Reasons To Reject:**

- The paper lacks several details about the data generation process (elaborated in questions), making it extremely hard to judge the correctness and scope of the generated data, its usefulness in future tasks, and the generalizability of the fine-tuned models.
- The primary methodology of the paper is relatively straightforward and well adopted elsewhere (generate synthetic training data with a multi-billion parameter LLM, filter, fine-tune smaller LLM) - leaving the data contributed and the fine-tuned models being the primary contributions.

**Reproducibility:**

2: Would be hard pressed to reproduce the results. The contribution depends on data that are simply not available outside the author's institution or consortium; not enough details are provided.

**Reviewer Confidence:**

4: Quite sure. I tried to check the important points carefully. It's unlikely, though conceivable, that I missed something that should affect my ratings.

**Typos Grammar Style And Presentation Improvements:**

- MatBERT outperforms the HoneyBee model on paragraph classification micro-F1; please fix the highlighting in Table 3.

---

> ### Author Rebuttal · Authors · 2023-08-29
>
> We thank you for feedback and are happy that you recognize our contribution in iterative instruction generation for training HoneyBee, the first materials science LLM. We also appreciate that you noticed that our method provided strong empirical gains over prior methods and that the paper was easy to read.  To clarify, the general process for creating instructions-data and training HoneyBee is:
>
> **Instruction Generation and Finetuning:**
>
> 1. Data Generation: ChatGPT creates training data (data_train0 [1-10]).
> 2. Verification: Claude removes low-scored data (data_train0 [1 and 2]).
> 3. Fine-Tuning: LLAMA-7b becomes HoneyBee-7b-stage-1 with data (data_train0 [3-10]).
>
> **Evaluation:**
>
> * Evaluation: HoneyBee-7b-stage-1 performs inference on new test data (data_test0), crafted by ChatGPT, with outputs evaluated by GPT-4.
>
> **Feedback Response:**
>
> * Response Generation: HoneyBee-7b-stage-1 generates responses for the test data (data_test0).
> * Scoring Responses: GPT-4 spots weak responses (data_test0[7 and 8]).
>
> **Instruction Adaptation and Improvement:**
> * Focusing on Weaknesses: ChatGPT crafts more training (data_train1) and test data (data_test1), focusing on issues identified by GPT-4.
> * Fine-Tuning Stage 2: HoneyBee-7b-stage-1 refines to HoneyBee-7b-stage-2.
> * Re-Evaluation: HoneyBee-7b-stage-2 is tested with new test data (data_test1).
>
> **Feedback Loop:**
> * Iterative Process: This loop continues until HoneyBee-7b reaches the desired performance.
>
> We include relevant prompt details used for instruction generation, verification and evaluation in the Appendix along with the full prompt data available in our code and dataset, which will be released upon publication. Next we address some of the feedback and questions you asked:
>
> > Reasons to Reject 1: Data generalization details.
>
> Please see the answers to the specific questions for more details.
>
> > Reasons to Reject 2: The primary methodology of the paper is relatively straightforward and well adopted elsewhere (generate synthetic training data with a multi-billion parameter LLM, filter, fine-tune smaller LLM) - leaving the data contributed and the fine-tuned models being the primary contributions.
>
> It is well known that LLMs suffer from hallucination problems. When the text generated by the model does not follow the original text (Faithfulness) or does not conform to the facts (Factualness), we can consider that the model exhibits hallucination problems. Therefore, our contribution also lies in proposing an iterative fine-tuning framework that uses multiple steps to eliminate low-quality, potentially hallucinated, fine-tuning data and obtain trustworthy models through progressive training, which is fundamentally different from existing instruction-tuning approaches.
>
> > Question A:
>
> Yes, the "instruction" and "input" are formatted with a prompt template to ask ChatGPT to generate relevant data with the generated data being referred to as “instruction data”. We will revise our manuscript to make this clearer.
>
> > Question B: 1) I see that Table 5 lists the 20 tasks, where were these tasks obtained? i.e is it an existing dataset? 2) What do the second columns of Table 4 and 5 mean? 3) Where was the input fed to ChatGPT for each of the tasks obtained? 4) Was a single instruction per task used for the 20 tasks? 5) Were the instructions manually authored?
>
> 1）Initially, we asked ChatGPT to generate 20 candidate tasks which we then downselected to the most appropriate NLP tasks related to materials science. The tasks are not directly related to an existing dataset, but were informed by a review of current relevant literature.
>
> 2）The second column represents the number of instructions. We will add this to the Table captions.
>
> 3)-5) As we clarify in the first part of our response, we use a manually designed prompt template to generate instruction data for which we only need to specify the input text and task when generating instruction data. The input text is randomly selected from a span of a materials science academic paper from arXiv, and the task is randomly chosen from 20 candidates. Together these templated prompts, which we describe in more detail in the Appendix, provide a scalable and repeatable manner to generate high-quality, relevant instructions data.
>
> > Question C: Was there an overlap in the input data to ChatGPT and the MatSci-NLP tasks?
>
> No, we kept the data separate for independent evaluation of the performance of HoneyBee and other baseline models.
>
> > Question D: Did ChatGPT, Claude, or GPT4 use any few shot examples in their prompts?
>
> No, these models already have very good zero-shot capabilities and do not require providing additional examples. Only when generating new instructions-data based on the feedback from the previous stage, we will select bad cases to put into the prompt so that the instructor can improve based on previous instructions and thereby generate higher-quality data.
>
> > Question E: Sec 3.1.2: "The accuracy of the instruction data is evaluated by comparing it with known facts or credible sources" - where are the facts and credible sources obtained from?
>
> We specify in the verification and evaluation prompts: "evaluate the instruction data by comparing it with known facts or credible sources". As such, known facts or credible sources come from the knowledge of the large model itself. Generally speaking, ChatGPT, Claude, and GPT-4 are trained with massive samples and themselves constitute a huge knowledge base. In future work, we aim to have these facts be based on external knowledge which would make instruction verification more robust.
>
> > Question F:
>
> Here the term "instructions" refers to the initial output of ChatGPT (the instructor) for finetuning HoneyBee. Concretely, we use a prompt to ask chatGPT to generate instructions-data (question-answering pairs). In the case of instruction refinement, the prompt includes previous instructions along with feedback from the evaluator model (GPT-4). We will revise our manuscript to make this clearer.
>
> > Question G:
>
> In our experiments, we follow the same low-resource procedure described in Song et al. (2023) by splitting the data in MatSci-NLP into 1% training subset and a 99% testing subset for evaluation. After that, all models, including HoneyBee, are finetuned on the training subset and evaluated on the test subset based on MatSciNLP data.
>
> > Question H:
>
>  Thanks for the feedback. We have performed zero-shot experiments for Chat-GPT, Claude and GPT-4 for Table 3 (shown in the table below), which we plan to include the results in the camera-ready version if the paper is accepted.
>
> **Zero-shot setting on MatSci-NLP Benchmark:** Data shown are Micro-f1/macro-f1 for each task
>
> | Task | NER | Relation Extraction | Event Argument Extraction | Paragraph Classification | Synthesis Action Retrieval | Sentence Classification | Slot filling | Overall |
> | ------ | ------ | ------ | ------ | ------ | ------ | ------ | ------ | ------ |
>
> ChatGPT |  0.063/0.052 | 0.232/0.145 | 0.204/0.203 | 0.433/0.450 | 0.300/0.183 | 0.320/0.318 | 0.368/0.280 | 0.274/0.233 |
>
> Claude | 0.063/0.048 | 0.232/0.143 | 0.195/0.169 | 0.442/0.467 | 0.280/0.177 | 0.329/0.326 | 0.393/0.305 | 0.276/0.234 |
>
> GPT-4 | 0.189/0.121 | 0.445/0.432 | 0.453/0.353 | 0.679/0.522 | 0.743/0.677 | 0.788/0.689 | 0.502/0.483 | 0.543/0.468 |
>
> From the results above, we can see that Honey-7b and HoneyBee-13b outperform both ChatGPT and Claude and are generally competitive with GPT-4. HoneyBee is also an open-source LLM and therefore more accessible compared to all the closed-source models (ChatGPT, Claude, GPT-4).

---

### Official Review · Reviewer_1nqS · 2023-08-11

**Typos Grammar Style And Presentation Improvements:** Figure 2 caption
**Soundness:** 3

**Excitement:**

3: Ambivalent: It has merits (e.g., it reports state-of-the-art results, the idea is nice), but there are key weaknesses (e.g., it describes incremental work), and it can significantly benefit from another round of revision. However, I won't object to accepting it if my co-reviewers champion it.

**Paper Topic And Main Contributions:**

Summary: The paper introduces a framework, MatSci-Instruct, designed for data generation in materials science. This process is then employed to fine-tune a large language model, named HoneyBee, making it specialized for materials science tasks. In addition, the authors use three different LLMs as instructor, verifier, and evaluator, respectively, to assist in this entire process of data generation and student model training. Experimentally, HoneyBee is evaluated using the MatSci-NLP benchmark, where it shows superiority over existing language models for materials science tasks.

Main Contributions:
1. Introduction of MatSci-Instruct: A LLM-assisted framework for data curation in materials science. This process uses one model to instruct and generate data and another separate model to verify its accuracy and relevance.
2. Development of HoneyBee: HoneyBee is the first billion-parameter language model that is specialized for materials science tasks.

**Questions For The Authors:**

See "reasons to reject". In addition, considering that the authors utilize 3 different LLMs (ChatGPT, Claude, GPT-4) to aid in the whole process, perhaps it would be useful to list all of their results in Table 2 as a reference.

**Reasons To Accept:**

1. The first distilled data in the field of materials science, and the authors verify the validity of the data.
2. The first instruction-tuned model specific to material science, which is based on LLaMA and LoRA.

**Reasons To Reject:**

1. Poor presentation of the proposed method. From the main text of this paper, I personally cannot understand how the evaluator module works. What data does GPT-4 evaluate HoneyBee's output on, the full set of data generated in the current iteration, or a subset? Why do we need to train HoneyBee first, then evaluate its output, and not just evaluate the original generated data? What is the motivation or benefit of the proposed method in the paper as opposed to directly using GPT-4 and Claude as two different verifiers to filter and revise the data?
2. Lack of necessary details in the experiment. For example, I'm curious how the training and test set are divided. Is the test set also auto-generated, and if so, does this introduce specific bias?

**Reproducibility:**

2: Would be hard pressed to reproduce the results. The contribution depends on data that are simply not available outside the author's institution or consortium; not enough details are provided.

**Reviewer Confidence:**

3: Pretty sure, but there's a chance I missed something. Although I have a good feel for this area in general, I did not carefully check the paper's details, e.g., the math, experimental design, or novelty.

---

> ### Author Rebuttal · Authors · 2023-08-29
>
> We thank you for your feedback and are happy to hear that you recognize our contribution with HoneyBee as the first instructions-finetuned materials science language model with a trustworthy, iterative instruction generation process leveraging different LLMs. To clarify, the general process for creating instructions-data and training HoneyBee is:
>
> **Instruction Generation and Finetuning:**
>
> 1. Data Generation: ChatGPT creates training data (data_train0 [1-10]).
> 2. Verification: Claude removes low-scored data (data_train0 [1 and 2]).
> 3. Fine-Tuning: LLAMA-7b becomes HoneyBee-7b-stage-1 with data (data_train0 [3-10]).
>
> **Evaluation:**
>
> * Evaluation: HoneyBee-7b-stage-1 performs inference on new test data (data_test0), crafted by ChatGPT, with outputs evaluated by GPT-4.
>
> **Feedback Response:**
>
> * Response Generation: HoneyBee-7b-stage-1 generates responses for the test data (data_test0).
> * Scoring Responses: GPT-4 spots weak responses (data_test0[7 and 8]).
>
> **Instruction Adaptation and Improvement:**
> * Focusing on Weaknesses: ChatGPT crafts more training (data_train1) and test data (data_test1), focusing on issues identified by GPT-4.
> * Fine-Tuning Stage 2: HoneyBee-7b-stage-1 refines to HoneyBee-7b-stage-2.
> * Re-Evaluation: HoneyBee-7b-stage-2 is tested with new test data (data_test1).
>
> **Feedback Loop:**
> * Iterative Process: This loop continues until HoneyBee-7b reaches the desired performance.
>
> We include relevant prompt details used for instruction generation, verification and evaluation in the Appendix along with the full prompt data available in our code and dataset, which will be released upon publication. Next we address some of the feedback and questions you asked:
>
> > What data does GPT-4 evaluate HoneyBee's output on, the full set of data generated in the current iteration, or a subset?
>
> We use structured prompts to ask GPT-4 to evaluate the output of HoneyBee based on the evaluation standards described in Sec 3.1.2. At each stage, the Evaluator (GPT-4) will evaluate the full set of HoneyBee’s outputs on a test set based on a standard set of evaluation prompts that we plan to publish along with the code and data.
>
> > Why do we need to train HoneyBee first, then evaluate its output, and not just evaluate the original generated data?
>
> Our framework includes an iterative loop that iteratively improves the quality of the instructions and model performance as shown in Table 2 and Table 3 respectively. The feedback improves HoneyBee’s underlying modeling ability compared to the previous iterations as well as the instructor, verifier and evaluator models. Moreover, HoneyBee is an open-source model that will be released to the community at no cost, making it more accessible than closed-source LLMs.
>
> > What is the motivation or benefit of the proposed method in the paper as opposed to directly using GPT-4 and Claude as two different verifiers to filter and revise the data?
>
> If we had GPT-4 and Claude as two consecutive verifiers, the verified data would contain a greater bias compounded by both of the models. By keeping the evaluation independent from the verification we reduce such bias and also enable the iterative refinement of instructions based on the performance of the HoneyBee model during each stage of training, which significantly improves performance as shown in Table 2 and Table 3. If GPT-4 and Claude were intermingled as verifiers, it would be more challenging to perform this iterative instruction refinement given the greater bias in the data used for finetuning.
>
> > Q：Experiment details and train/test data set generation?
>
> The details and hyperparameters of the experiment are shown in Appendix A. The test set is auto-generated with the same generation process as the train set but only introduced during evaluation of modeling performance, which makes model evaluation more robust. Additionally, we use a public dataset MatSci-Bench to test the ability of model generalization to realistic NLP applications shown in Section 4.3. The zero-shot generalization abilities shown in Table 3 make our results pertaining to HoneyBee’s materials science knowledge particularly convincing.
>
> > Q: list results of ChatGPT, Claude, and GPT-4 in Table 2".
>
> Claude and GPT-4 are used as a verifier and an evaluator, and hence are responsible for evaluating the performance of different models. Therefore, we cannot use them as baselines for comparison in Table 2 given that the results are based on their own output.

---

### Meta-Review · Area_Chair_5fYo · 2023-09-23

**Recommendation:** 3

**Metareview:**

This paper describes a methodology for instruction finetuning an LLM for the domain of materials science, including the release of a new benchmark for this domain. While reviewers agreed that the application area and new benchmark were interesting, they weren't very excited about the novelty of the approach itself compared with previous work, and felt that the paper was in some places unclear or that details were missing. The authors should do a careful pass to revise the manuscript for clarity and inclusion of details following the reviewers' feedback in subsequent drafts.

---

### Decision · Program_Chairs · 2023-10-07

**Decision:**

Accept-Findings

**Comment:**

This paper describes a methodology for instruction finetuning an LLM for the domain of materials science, including the release of a new benchmark for this domain. While reviewers agreed that the application area and new benchmark were interesting, they weren't very excited about the novelty of the approach itself compared with previous work, and felt that the paper was in some places unclear or that details were missing. The authors should do a careful pass to revise the manuscript for clarity and inclusion of details following the reviewers' feedback in subsequent drafts.